# Increased Potential of Bone Formation with the Intravenous Injection of a Parathyroid Hormone-Related Protein Minicircle DNA Vector

**DOI:** 10.3390/ijms22169069

**Published:** 2021-08-23

**Authors:** Jang-Woon Kim, Narae Park, Jaewoo Kang, Yena Kim, Hyerin Jung, Yeri Alice Rim, Ji Hyeon Ju

**Affiliations:** 1Catholic iPSC Research Center (CiRC), CiSTEM Laboratory, College of Medicine, The Catholic University of Korea, Seoul 06591, Korea; kimjw2108@hanmail.net (J.-W.K.); narae5322@gmail.com (N.P.); tiq2005@naver.com (J.K.); kyena0430@gmail.com (Y.K.); ilovehyelin@gmail.com (H.J.); llyerill0114@gmail.com (Y.A.R.); 2Department of Biomedicine & Health Science, Seoul St. Mary’s Hospital, College of Medicine, The Catholic University of Korea, Seoul 07345, Korea; 3Department of Internal Medicine, Division of Rheumatology, Seoul St. Mary’s Hospital, College of Medicine, The Catholic University of Korea, Seoul 06591, Korea

**Keywords:** osteoporosis, ovariectomized mice, minicircle DNA vector, parathyroid hormone-related protein, trabecular bone structure

## Abstract

Osteoporosis is commonly treated via the long-term usage of anti-osteoporotic agents; however, poor drug compliance and undesirable side effects limit their treatment efficacy. The parathyroid hormone-related protein (PTHrP) is essential for normal bone formation and remodeling; thus, may be used as an anti-osteoporotic agent. Here, we developed a platform for the delivery of a single peptide composed of two regions of the PTHrP protein (1–34 and 107–139); mcPTHrP 1–34+107–139 using a minicircle vector. We also transfected mcPTHrP 1–34+107–139 into human mesenchymal stem cells (MSCs) and generated Thru 1–34+107–139-producing engineered MSCs (eMSCs) as an alternative delivery system. Osteoporosis was induced in 12-week-old C57BL/6 female mice via ovariectomy. The ovariectomized (OVX) mice were then treated with the two systems; (1) mcPTHrP 1–34+107–139 was intravenously administered three times (once per week); (2) eMSCs were intraperitoneally administered twice (on weeks four and six). Compared with the control OVX mice, the mcPTHrP 1–34+107–139-treated group showed better trabecular bone structure quality, increased bone formation, and decreased bone resorption. Similar results were observed in the eMSCs-treated OVX mice. Altogether, these results provide experimental evidence to support the potential of delivering PTHrP 1–34+107–139 using the minicircle technology for the treatment of osteoporosis.

## 1. Introduction

Osteoporosis is a bone disease characterized by the loss of bone strength that leads to fragility fractures, which are accompanied by a decrease in the thickness of trabecular bones [1,2]. There are the following three categories of osteoporosis: (1) primary (estrogen deficiency- and age-associated), (2) secondary (of endocrine and metabolic origin; can also be drug-induced or related with genetic disorders), and (3) idiopathic (in young adults; of unknown origin) [3]. Estrogen deficiency during menopause impairs normal bone metabolism, with an increase in the osteoclastic bone resorption without a corresponding increase in the osteoblastic activity; this process, which involves bone remodeling, is named uncoupling. Thus, in addition to the reduction in trabecular bone thickness, the trabecular bone plate also changes to a rod-like structure, which increases the risk of osteoporosis fractures [2]. Currently, postmenopausal osteoporosis treatments are predominantly US Food and Drug Administration (FDA)-approved drug-based therapeutic agents that are categorized into the following two classes: anti-resorptive agents and anabolic agents.

The anti-resorptive agents were deemed effective in clinical trials, with an increase in the bone mineral density (BMD) and decreased fracture risk [4,5]. Among them are estrogen, Selective Estrogen Receptor Modulators (SERMs), calcitonin, and bisphosphonates. However, recent studies have reported that the long-term usage of anti-resorptive agents can induce negative effects such as the excessive suppression of physiological bone turnover, osteonecrosis, and atypical femoral fractures [4,5,6]. On the other hand, anabolic agents increase bone formation via osteoblast-mediated bone metabolism [7,8]. Among these agents are the parathyroid hormone analogs and PTH-related peptide (PTHrP) analogs including teriparatide and abaloparatide, which are both FDA-approved [9,10,11,12,13,14]. The protein structure of PTHrP includes the PTH-like N-terminal PTHrP peptide and the C-terminal PTHrP peptide. The PTHrP-null mice show a form of skeletal haploinsufficiency characterized by decreased bone volume [15]. Abaloparatide is a PTHrP analog drug that is used for the treatment of osteoporosis. The administration of a high dosage of abaloparatide on ovariectomized (OVX) rats for 6 weeks was associated with the improved trabecular microarchitecture and increased BMD, trabecular thickness (Tb.Th), and structure model index (SMI) [16]. In fact, the abaloparatide-treated (40 μg/kg) group showed a higher trabecular bone formation rate (BFR) in the lumbar spine (L5) than that observed in the groups treated with teriparatide in the OVX mice [17]. Similarly, the daily administration of PTHrP 1–36 and PTHrP 107–139 to OVX animals resulted in the same osteogenic effects. Additionally, PTHrP 107–139 inhibited the expression of *DKK1* and *Sost* in osteocytes and decreased the levels of bone resorption markers in the OVX mice [18]. Another study reported that recombinant hPTHrP 1–84 was more effective than hPTHrP 1–34 in the context of both renal calcium reabsorption and the stimulation of bone formation in the OVX mice [19]. However, despite these advantages, bone anabolic agents can also induce adverse effects, including hypercalcemia and dizziness [20]. The transplantation of mesenchymal stem cells (MSCs) also has the potential to enhance osteogenic differentiation, to increase the BMD in the context of osteoporosis. However, optimization is still needed to improve the bone-targeted efficacy of transplanted MSCs, and to associate this strategy with newly developed drugs or gene therapies for postmenopausal osteoporosis [21,22].

There are the following two strategies available for the delivery of genes: viral-based and non-viral-based ones. Viral-based systems are associated with a high efficiency of transfection in vivo. However, they have serious disadvantages such as immunogenicity and the consequent induction of inflammatory responses in vivo. On the other hand, non-viral-based systems usually consist of plasmid DNA carrying the gene of interest and show great promise as a safe in vivo gene transfer strategy [23,24]. A nontoxic delivery vector that demonstrates preclinical safety is crucial for the development of a gene therapy-based treatment for osteoporosis and minicircle vectors are thought as a good option. Due to the elimination of the bacterial backbone and the antibiotic resistance gene, minicircles have a relatively smaller size compared to other commercial vectors. The small size of minicircles increase the delivery rate of the gene-of-interest in vivo; thus, has great potential in gene therapy [23,25]. We have previously highlighted minicircles as an alternative option as an in vitro and in vivo delivery system for biologics, which usually have a high cost of development and production. Additionally, we confirmed that drug-encoding minicircles injected into collagen-induced arthritic mice resulted in drug synthesis [26,27,28,29]. However, the drug delivery by minicircles was never used in the context of osteoporosis in animal models.

In this study, gene therapy using minicircles for the treatment of osteoporosis was attempted as a proof-of-concept in the OVX mice. The therapeutic effect of one entire peptide composed of two regions of PTHrP (1–34 and 107–139) was confirmed through both the direct injection of minicircles and its delivery using transfected MSCs. The combination of the increased bone formation induced by PTHrP 1–34 and the decreased bone resorption induced by PTHrP 107–139 is expected to induce bone remodeling in vivo. The increased bone formation in the OVX mice demonstrates a successful in vivo delivery of the developed PTHrP-containing minicircles and supported its development as a new gene-therapy-based strategy for the treatment of osteoporosis.

## 2. Results

### 2.1. Generation of the mcPTHrP 1–34+107–139 Vector In Vitro

The PTHrP 1–34+107–139-encoding parental plasmid was designed and developed as shown in (Figure 1A). The bacterial backbone was removed from the parental vector by arabinose treatment, and minicircles (mcPTHrP 1–34+107–139) were generated. Double digestion with Bam HI and XbaI confirmed the accurate insertion of the cloned PTHrP 1–34+107–139 sequence (Figure 1B). Additionally, the green fluorescent protein (GFP) observed in HEK293T cells confirmed the expression of mcPTHrP 1–34+107–139 48 h post-transfection (Figure 1C). The co-expression of the PTHrP antibody and GFP was observed in mcPTHrP 1–34+107–139-transfected cells (Figure 1D). Reverse transcription-polymerase chain reaction (RT-PCR) analysis further confirmed the expression of PTHrP 1–34+107–139 in HEK293T cells 48 h post-transfection (Figure 1E). Due to the small size of the insert (282 bp), we also confirmed the potential false-positives caused by vector contamination in 48 h post-transfected cells by performing RT-PCR without the presence of reverse transcriptase (Appendix A). PTHrP 1–34+107–139 false-positive signals were not found in the HEK 293T cells and the mock group, indicating that the detected bands were the PTHrP 1–34+107–139 inserts. Taken all together, we confirmed the successful production of mc PTHrP 1–34+107–139.

### 2.2. In Vivo Detection of mcPTHrP 1–34+107–139

Mice were injected with 40 μg/kg of mcPTHrP 1–34+107–139 diluted in phosphate-buffered saline (PBS) three times from weeks 4–6 post-ovariectomy. The expression of PTHrP 1–34+107–139 was observed at protein levels in the spleen, kidney, and liver samples that were collected from the mice on day 3, 7, 15, and 37 after injection (Figure 2A–C). The gene levels were also confirmed in the same samples (Figure 2E–G). Through the positive expression of GFP and red fluorescence protein (RFP), we confirmed the successful in vivo delivery of mcPTHrP 1–34+107–139. In the spleen, the PTHrP and 1–34+107–139 protein expression was detected at 15 and 37 days post-injection (Figure 2A,E). The co-localization of GFP (green) and PTHrP signals (red) was clearly observed under high-power (×1000) magnification (Figure 2D). In the spleen, the PTHrP protein expression was the highest at day 15 post-injection (Figure 2A). The same was true in the context of kidney tissues, at both the protein and transcriptional levels (Figure 2B,D). Again, the co-localization of GFP and PTHrP signals (green and red, respectively) was confirmed (Figure 2D). In contrast, the expression of PTHrP 1–34+107–139 at the protein and transcriptional levels increased rapidly in the liver, peaking at day seven post-transfection (Figure 2C,D). We also confirmed if there are any false positive signals caused by vector contamination in the tissue samples as well (Appendix A). While false positive signals were observed in several samples; however, these results were negligible as the band intensity was higher in the results of transcriptase including samples. These results suggest that the mcPTHrP 1–34+107–139 was successfully delivered in vivo.

### 2.3. mcPTHrP 1–34+107–139 Promotes Bone Formation and Prevents Bone Resorption in OVX Mice with a Positive Impact on the Microarchitecture of Trabecular Bones

Next, to investigate the in vivo effects of mcPTHrP 1–34+107–139, the same administration protocol as above was used, followed by the evaluation of the microarchitecture of trabecular bones as well as of bone formation and resorption (Figure 3A). Four weeks post-mc injection, the bone microarchitecture of the distal femur (0.5–2 mm) was evaluated using micro-CT (Figure 3B) and reconstruction programs (Figure 3C–H). The bone fraction (BV/TV) was higher in the mcPTHrP 1–34+107–139-administered versus the control OVX mice, although the difference was not statistically significant (Figure 3C). Compared with the trabecular bone numbers (Tb.N) of the OVX control mice, those of the mcPTHrP 1–34+107–139-administered mice did not increase (Figure 3D). However, the trabecular thickness (Tb.Th) was significantly higher in the mcPTHrP 1–34+107–139-administered versus the control OVX mice (Figure 3E). Conversely, the trabecular bone pattern factor (Tb.Pf) was significantly lower in the mcPTHrP 1–34+107–139-administered versus the control OVX mice (Figure 3F). Consequently, the bone specific surface (BS/BV) (especially the trabecular bone density) (Figure 3G) and the structural model index of the trabecular bone (Figure 3H) were significantly decreased in the mcPTHrP 1–34+107–139-administered versus the control OVX mice. Additionally, we also determined the levels of two bone turnover markers, N-terminal propeptide of type 1 procollagen (PINP) (bone formation), and C-terminal telopeptide of type 1 collagen (CTX-1) (bone resorption) in the serum of mice 10 weeks post-ovariectomy. The PINP serum levels, significantly higher in the mcPTHrP 1–34+107–139-administered versus the control OVX mice suggested an increase in bone formation after the administration of mcPTHrP 1–34+107–139 (Figure 3I). In parallel, the significantly lower serum levels of CTX-1 in the mcPTHrP 1–34+107–139-administered versus the control OVX mice indicated a decrease in bone resorption after the administration of mcPTHrP 1–34+107–139 (Figure 3J).

### 2.4. Therapeutic Application of mcPTHrP 1–34+107–139-Transfected MSCs in the Context of OVX Mice

As the mcPTHrP 1–34+107–139 functionality was verified in HEK 293T cells, the same vector was transfected into MSCs via microporation using the Neon transfection system to generate engineered MSCs (eMSCs) (Appendix A). To investigate the effects of eMSCs in vivo, the OVX mice were intraperitoneally injected twice with MSCs and eMSCs (weeks four and six) (Figure 4A). The eMSCs expressed GFP 48 h after transfection (Figure 4B) and retained the MSCs phenotype by exhibiting a negative expression of CD45 and CD34 and a positive expression of CD73 and CD105. The MSCs marker expressions were similar in MSCs (passage 5) and eMSCs (Figure 4C). Altogether, these results indicate that eMSCs were successfully generated. Next, we used them to treat the OVX mice and looked once again at the bone parameters 10 weeks post-ovariectomy-micro-CT (Figure 4D) with the reconstruction method (Figure 4E–H). The bone fraction (BV/TV) was higher in the eMSCs-treated versus the control OVX mice, although the difference was not statistically significant (Figure 4E). As before, the Tb.N values were similar to the eMSCs-treated versus the control OVX mice at 10 weeks post-ovariectomy (Figure 4F). On the other hand, Tb.Th was higher, while the SMI of the trabecular bone was lower in the eMSCs-treated versus both the MSCs-treated and the control OVX mice (Figure 4G,H). The levels of PINP were significantly higher (Figure 4I) and those of CTX-1 were significantly lower (Figure 4J) in the eMSCs-treated versus both the MSCs-treated and the control OVX mice, suggesting an increased bone formation and a decreased bone resorption in the context of eMSCs treatment.

### 2.5. Protein Expression of PTHrP and PTH/PTHrP-R, and Bone Formation/Resorption in mcPTHrP 1–34+107–139-Administered Mice

We also measured the protein expression of PTHrP and PTH/PTHrP-R in the femurs of the mcPTHrP 1–34+107–139-administered and the control OVX mice, 10 weeks post-OVX. As expected, the protein levels of PTHrP were significantly higher in the mcPTHrP 1–34+107–139-administered versus the control OVX mice (Figure 5A). Similarly, the protein levels of PTH/PTHrP-R were also higher in the mcPTHrP 1–34+107–139-administered versus the control OVX mice, although the difference was not statistically significant (Figure 5B). The protein expression of osteocalcin protein was higher in the femur tissues of the mcPTHrP 1–34+107–139-administered versus the control OVX mice (Figure 5C), while that of TRAP protein expression was decreased in mice administered mcPTHrP 1–34+107–139 (versus the control OVX mice) (Figure 5D). These differences were not statistically significant.

### 2.6. Protein Expression of PTHrP and PTH/PTHrP-R, and Bone Formation/Resorption in eMSCs-Treated Mice

Similarly, the protein expression of PTHrP and PTH/PTHrP-R were also performed in the femur tissues of MSCs/eMSCs-treated mice, 10 weeks post-ovariectomy. The PTHrP protein expression was higher in both the MSCs- and eMSCs-treated versus the control OVX mice (Figure 6A). Similarly, the PTH/PTHrP-R protein expression was also higher in both the MSCs- and eMSCs-treated versus the control OVX mice; however, these differences were not statistically significant (Figure 6B). Additionally, no major differences in the osteocalcin protein levels were detected in the femur tissues of the different groups (Figure 6C). The TRAP protein expression was significantly decreased in both the MSCs and eMSCs-treated versus the control OVX mice (Figure 6D); no significant differences were detected between the MSCs- and eMSCs-treated animals.

## 3. Discussion

PTHrP is expressed by cells in the early osteoblastic lineage, suggesting a role for PTHrP in bone cell regulation. In this study, we generated a vector encoding the nucleotide sequence of PTHrP 1–34 (involved in bone formation) combined with 107–139 (for osteoclast inhibition) and developed a new method to deliver this therapeutic protein in the OVX mice using minicircles. Our previous studies reported that minicircles allow the expression of genes for a longer duration than that in the context of other conventional vectors, without the need for integration into the host genome [25,27,28,29]. Remarkably, in this study we show an increased expression of the PTHrP 1–34+107–139 gene sequence in the spleen and kidneys of mice 15 days after the injection of mcPTHrP 1–34+107–139. On the other hand, the expression of PTHrP 1–34+107–139 expression was higher on day 7 and slightly lower on days 15 and 37 in the liver, which could be explained by the initial migration of mcPTHrP 1–34+107–139, and subsequent PTHrP expression. We also showed the applicability of this vector in vivo. Ten weeks post-ovariectomy, the protein expression of PTHrP and PTH/PTHrP-R was higher in the femur tissues of the OVX mice treated with mcPTHrP 1–34+107–139 or eMSCs. Based on these results, the bioactivity of the mcPTHrP 1–34+107–139 vector was confirmed in the femur tissues.

An ovariectomy is known to decrease parameters including the BV/TV, Tb.N, and Tb.Th in mice, and increase others including the Tb.Sp and SMI over several weeks [30]. Several groups have highlighted PTHrP as a potent endogenous bone anabolic agent in OVX mice [8,9]. For instance, the intermittent administration of PTHrP 1–36 showed bone anabolic action in estrogen-depleted subjects, as suggested by the increase in the osteoblast and osteocyte numbers and by the trabecular microarchitecture. Similarly, PTHrP 107–139 showed an osteogenic capacity and the consequent anti-resorptive and anabolic features [8]. To our knowledge, after mc vector injection, three of the five mice in the mcPTHrP 1–34+107–139 group showed an increased bone mass (BV/TV) and trabeculae (Tb.N) compared with their control OVX counterparts. Nonetheless, the administration of mcPTHrP 1–34+107–139 significantly increased the Tb.Th, and decreased the BS/BV, Tb.Pf, and SMI (versus the control OVX mice), 10 weeks post-ovariectomy. The SMI and BS/BV values are influenced by morphological changes in the trabeculae and the transformation from plate-like to rod-like structures, and previously, the critical stress intensity factor values were correlated with the microarchitecture of the osteoporotic cancellous tissue [31,32].

Bone turnover occurs via CTX-1- and PINP-mediated bone resorption and formation, respectively [33]. The amount of bone resorption usually equals that of bone formation in every bone remodeling unit (remodeling balance). However, in the bone remodeling units after menopause, bone formation is reduced compared with bone resorption in the same remodeling cycle (remodeling imbalance). Therefore, increased concentrations of bone turnover markers may be associated with increased bone loss and fracture risk in postmenopausal women [20]. Our results clearly suggest that the administration of mcPTHrP 1–34+107–139 promotes bone formation and prevents bone resorption, as per the significant elevation of the serum levels of PINP and the significant reduction in the serum levels of CTX-1 in mcPTHrP 1–34+107–139-administered OVX mice 10 weeks post-ovariectomy (versus the control OVX mice).

A potential alternative for the delivery of biologics using MSCs transfected with TNFR2-encoding minicircles and cell-based therapy was previously confirmed in collagen-induced arthritis mice [16]. Additionally, previous preclinical research suggested that allogeneic MSC therapy could serve as a promising anabolic option in the management of glucocorticoid-induced osteoporosis programs (GIOP) [34]. To our knowledge, this is the first study to confirm the therapeutic application of eMSCs with the combination of PTHrP 1–34+107–139 in OVX mice. In fact, as per the micro-CT analysis, while the mice treated with MSCs and eMSCs showed no increase in the Tb.N compared with the OVX group mice, they showed an increased bone mass, together with increased serum levels of PINP (bone formation marker). Therefore, our results suggest that the injection of eMSCs as well as mcPTHrP 1–34+107–139 restored the remodeling balance in the OVX mice.

Several limitations could be improved in the future. First, only one dose and one administration scheme were tested. An evaluation of an increased dosage (80 µg/kg), of a different frequency of administration, and of better delivery routes (intra-bone marrow) must be carried out in the future. In the beginning, we had some difficulties comparing the injection of mc vectors and eMSCs cell counts. Second, the efficacy of the eMSCs was difficult to obtain with multiple injections. Further animal experiments based on the comparison of the injection of mc vectors, MSCs, and eMSCs must be performed. We need to compare the minicircle DNA vector and the plasmid DNA vector in further studies.

In conclusion, our results clearly suggest that the mcPTHrP 1–34+107–139 vector can be used for in vivo bone regeneration. Therefore, this proof-of-concept study highlights a new strategy for the development of a future regenerative approach for the treatment of osteoporosis.

## 4. Materials and Methods

### 4.1. Production of the Mc PTHrP 1–34+107–139 Vector

The mock parental plasmid was purchased from SBI (System Bioscience, Mountain View, CA, USA; # MN501A-1). The parental plasmid includes the kanamycin resistance gene and the apUC origin of replication, enveloped by the attP and attB sites, which are identified by φC31 integrase. The PTHrP 1–34+107–139 sequence was subcloned into the mock parental plasmid, pMC.CMV-MCS-EF1-GFP-SV40-Poly A (7063 bp); PTHrP 1–34+107–139 cDNA was inserted at the Bam HI and Xba I restriction sites in the multiple cloning site downstream to the CMV promoter. The sequence of PTHrP 1–34+107–139 is shown in Appendix A. The mc vector PTHrP 1–34+107–139 was then obtained as described previously [15,16]. Briefly, *Escherichia coli* ZYCY10P3S2T cells transformed with the mock and PTHrP 1–34+107–139 plasmids were grown overnight in Terrific broth (TB) containing 50 µg/mL kanamycin in a 37 °C shaking incubator. A single colony was then grown for 8 h in 2 mL of Luria-Bertani (LB) broth, also supplemented with kanamycin. Subsequently, cultures were mixed with LB broth containing 0.02% l-(+)- arabinose and incubated for 5 h. The mc DNA vector was isolated using the Nucleobond Xtra Midi kit (Macherey-Nagel, Duren, Germany; # 740410.100). The successful integration of PTHrP 1–34+107–139 was confirmed via gel imaging after Bam HI and Xba I restriction.

### 4.2. HEK 293T Cell Culture and MC Vector Transfection

Human embryonic kidney (HEK293T) cells were cultured in Dulbecco’s Modified Eagle Medium (DMEM; Thermo Fisher Scientific, Waltham, MA, USA; #11965-092) supplemented with 7.5% fetal bovine serum (FBS), 100 U/mL penicillin, and 100 µg/mL streptomycin (Thermo Fisher Scientific; #15240096). HEK293T cells were transfected with mc vectors (Mock, PTHrP 1–34+107–139) using the Lipofectamine 2000 reagent (Thermo Fisher Scientific; #11668019) following the manufacturer’s instructions. On the day before transfection, HEK293T cells (70% confluent cultures) were dissociated and seeded at 4.5 × 10^4^ cells per cm^2^ in DMEM without 7.5% FBS, 100 U/mL penicillin, and 100 µg/mL streptomycin; the cells were then incubated overnight. At 48 h post-transfection, the expression of green fluorescence protein (GFP) in the transfected HEK293T cells was measured via fluorescence microscopy (Carl Zeiss, Oberkochen, Germany).

### 4.3. Immunocytochemistry

We performed immunofluorescence staining 48 h after the seeding of transfected HEK 293T cells into culture plates containing poly-lysine-coated glass coverslips. The samples were fixed in 4% paraformaldehyde for 30 min at room temperature (RT), and the coated glass coverslips were washed in Tris-buffered saline with 0.01% Tween 20. The cells were then permeabilized using 0.1% Triton X-100 for 10 min at RT and blocked with PBS containing 2% bovine serum albumin (BSA; Sigma-Aldrich, St Louis, MO, USA) (PBA) for 30 min at RT. To analyze the expression of PTHrP in transfection cells, double-labeling studies were performed; anti-PTHrP primary antibodies were added for 2 h at RT (1:200 in 2% PBA; Novusbio, Barton Lane, Abingdon, UK; #3H1-5G8) and Alexa Fluor 594-conjugated secondary antibodies were added for 1 h at RT (1:200 in 2% PBA; Life Technologies, Carlsbad, CA, USA). Cells were finally washed and mounted using the antifade mounting reagent (Thermo Fisher Scientific) and observed under a fluorescence microscope (Carl Zeiss); positive cells were defined as those with the co-localization of DAPI (blue), GFP (green), and PTHrP (red) signals (low-powered ×50 magnification).

### 4.4. Transfection of Human MSCs with the Mc PTHrP 1–34+107–139 Vector

Bone marrow-derived human MSCs were purchased from The Catholic Institute of Cell Therapy, South Korea. MSCs were maintained in DMEM supplemented with 20% FBS, 100 U/mL penicillin, and 100 µg/mL streptomycin. At passage 5, MSCs (1 × 10^6^) were transfected with the vector via electroporation using the Neon transfection system (Thermo Fisher Scientific; #MPK5000S). Electroporation was performed with 1400 pulse voltage, 20 pulse width, and 2 pulses. The induced eMSCs were cultured in DMEM supplemented with 20% FBS, 100 U/mL penicillin, and 100 µg/mL streptomycin for 5 days.

### 4.5. Characterization of eMSCs

The cultured eMSCs and MSCs were subjected to flow cytometric analysis. In brief, aliquots of 1 × 10^6^ cells per mL were immunolabeled at RT for 30 min with the following antibodies: anti-human CD34, CD45, and CD73 antibodies conjugated to PerCP-Cy5.5 and anti-human CD105 antibodies conjugated to PE-Cyanine 7 (BD Biosciences, San Jose, CA, USA). Cells were then acquired using an LSR Fortessa cell analyzer (BD Biosciences). The data were analyzed using the FlowJo 7.6.5 software (TreeStar Inc., Ashland, OR, USA).

### 4.6. Animal Model and Group Allocation

Twelve-week-old C57BL/6 female mice (Orient Bio Inc., Seongnam, Korea) with a mass range of 17–23 g were OVX to induce postmenopausal osteoporosis. Briefly, the mice were anesthetized with isoflurane via inhalation, and their abdominal region was shaved and sterilized using antiseptic betadine. The ovaries were then exposed via a transverse ventral incision and removed. After ovariectomy, the muscle layer was sutured with absorbable sutures, and the skin was sutured with non-absorbable sutures. Postoperatively, mice were sterilized using antiseptic betadine, and 10 mg/kg of gentamicin and ketoprofen were administered for 3 days. All animal experimental procedures were reviewed and approved by the Animal Studies Committee of the School of Medicine, The Catholic University of Korea (IACUC approval No. CUMC-2017-0250-03). At 4 weeks post-ovariectomy, mice were randomized into the following groups: sham group (*n* = 5), OVX group (*n* = 8), OVX + mcPTHrP 1–34 +107–139 group (*n* = 8), OVX + MSCs group (*n* = 7), and OVX + eMSCs group (mcPTHrP 1–34+107–139 MSCs, *n* = 8).

### 4.7. In Vivo Delivery of mcPTHrP 1–34+107–139 via Intravenous Injection

Four weeks post-ovariectomy, the mcPTHrP 1–34+107–139 DNA vector was delivered hydrodynamically via intravenous injection into the tail vein. Mice were injected with 40 μg/kg of mcPTHrP 1–34+107–139 in 1.8 mL of PBS (three injections from weeks 4–6 post-ovariectomy). Tissue samples from the spleen, kidneys, and liver were collected on days 3, 7, 15, and 37 after injection.

### 4.8. In Vivo Delivery of eMSCs

Four weeks post-ovariectomy, MSCs and eMSCs (1 × 10^6^ cells) were resuspended in 150 µL of PBS. MSCs and eMSCs were injected twice (at weeks 4 and 6 post-ovariectomy) via intraperitoneal injection.

### 4.9. Reverse Transcription-Polymerase Chain Reaction

To confirm the gene expression of mcPTHrP 1–34+107–139, transfected HEK293T cells and tissue samples were homogenized on ice in TRIZol solution (Thermo Fisher Scientific). RNA samples (0.5 µg/µL) were extracted for the subsequent synthesis of first-strand cDNA following the manufacturer’s instruction (Thermo Fisher Scientific). The primer sequences used for the detection of PTHrP 1–34+107–139 were 5′-catgacaaggggaagtccat-3′ (forward), and 5′-gacgttgtggaggtgtcaga-3′ (reverse) (158 bp product). Of note, we excluded potential false-positive signals (vector contamination) using reactions without reverse transcriptase. Bands were quantified as the mean band intensity and normalized to that of GAPDH using the Image J software (version 1.44 m; NIH, Bethesda, MD, USA).

### 4.10. Western Blot

Ten weeks post-ovariectomy, the femur bones were extracted and snap-frozen with liquid nitrogen. The bones were then crushed under liquid nitrogen, and the bone powder was homogenized and incubated at 4 °C for 1 h in a tissue protein extraction reagent (Thermo Scientific) supplemented with 1 protease inhibitor cocktail tablet (Roche, Grenzacherstrasse, Basel, Switzerland) and 1 mM PMSF. The extracted protein amount was then quantified using the Bicinchoninic Acid (BCA) protein assay. Proteins (30 mg/mL) were separated using SDS-polyacrylamide gel electrophoresis and, subsequently, transferred onto a nitrocellulose blotting membrane (GE Healthcare, Freiburg, Germany) for 1 h at 4 °C. Membranes were blocked with 3% bovine serum albumin (BSA) for 1 h at RT, and, subsequently, incubated with the following primary antibodies overnight at 4 °C: anti-PTHrP (1:500; Novusbio), anti-PTH/PTHrP-R (1:500; Santa Cruz Biotechnology, Dallas, TX, USA), anti-osteocalcin (1:500; Santa Cruz Biotechnology), and anti-TRAP (1:500; Santa Cruz Biotechnology). The next day, membranes were washed and incubated with the respective secondary antibodies for 1 h at RT. The assessment of protein expression was performed using the ECL solution, followed by the exposure of the membrane using the ImageQuant LAS 4000 system (BioRad, Hercules, CA, USA). The quantification of the intensity of the protein bands was performed using the multi-gauge V 3.0 software (Fujifilm, Tokyo, Japan).

### 4.11. Immunofluorescence Staining

Spleen, kidney, and liver tissue samples were fixed overnight in 4% paraformaldehyde (PFA) and then incubated in 15% and 30% sucrose also overnight. Thereafter, the tissue samples were embedded in optimal cutting temperature (OCT, Tissue-Tek; Sakura Finetek USA, Torrance, CA, USA) embedding matrix compound and snap-frozen in liquid nitrogen. Tissues sections (5 μm thickness) were then obtained and mounted on slides. The loss of GFP signals was prevented via drying the cryosections followed by direct fixation with 4% PFA pre-warmed at 30–37 °C, as reported elsewhere [35,36]. The slides were rehydrated and incubated in boiling citrated buffer (Sigma Aldrich, St. Louis, MO, USA) for 30 min at 65 °C for antigen retrieval. Afterward, endogenous peroxidase activity was blocked with 3% hydrogen peroxide (Sigma Aldrich) for 10 min at RT. The slides were washed and blocked with TBS containing 1% BSA for 1 h at RT. To identify PTHrP expression in mcPTHrP 1–34+107–139, double-labeling studies were performed using a primary antibody to the PTHrP (1:200, Novusbio). The next day, for fluorescence staining, the Alexa Fluor 594 goat anti-mouse IgG (H+L) antibody (1:200) was used. Secondary antibody was washed and DAPI staining for 10 min at RT. Slides were washed and mounted using antifade mounting reagent (Thermo Fisher Scientific). The stained slides were observed under a high-powered magnification using a confocal microscope (LSM 7000, Carl Zeiss).

### 4.12. Micro CT Analysis

Ten weeks post-ovariectomy, the distal femurs were imaged at a scanning voxel size of 13.85 µm with a high-resolution microtomographic system (Sky Scan 1173, ver. 1.7.0.4; Brucker, Billerica, MA, USA). Using the SkyScan reconstruction program, bone microarchitectural parameters in the trabecular bone regions were calculated to determine the bone volume/total volume fraction (BV/TV), the trabecular number (Tb.N), and the trabecular-specific surface area (BS/BV) based on the trabecular bone microarchitecture/trabecular thickness (Tb.Th), trabecular bone pattern factor (Tb.Pf), and structure model index (SMI).

### 4.13. Quantification of the Serum Bone Turnover Markers

Blood samples were collected 10-weeks post-ovariectomy and allowed to coagulate. The serum levels of the bone formation marker PINP and the bone resorption marker CTX-1 were then measured using commercial enzyme immunoassays (EIA), according to the manufacturers’ instructions: Rat/Mouse PINP EIA (Immunodiagnostic Systems, East Boldon, South Tyneside, UK; AC-33F1) and RatLaps^TM^ EIA (Immunodiagnostic Systems; # AC-06F1), respectively.

### 4.14. Statistical Analysis

All results are expressed as the mean ± standard error of the mean (SEM). Statistical relevance was determined using the Kruskal–Wallis test followed by the Fisher’s Least Significant Difference (LSD) (†) post hoc test for multiple comparisons, with the aid of the SPSS program (IBM Corporation, Armonk, New York, USA). Inter-group comparisons were also performed using the Mann–Whitney (*) when the Kruskal–Wallis test was deemed significant. A *p*-value of less than 0.05 was considered statistically significant (*† *p* < 0.05, **†† *p* < 0.01, ***††† *p* < 0.001, n.s. = not significant).

## 5. Conclusions

We show that the mcPTHrP 1–34+107–139 efficiently increases the quality of the trabecular bone structure and promotes new bone formation in vivo. In parallel, mcPTHrP 1–34+107–139 also prevents new bone resorption in the bone remodeling cycle of the OVX mice. Although this strategy is at the proof-of-concept stage, the use of mcPTHrP 1–34+107–139 and eMSCs (mcPTHrP 1–34+107–139 MSCs) represents a potential alternative method for the delivery of biologics for gene- and cell-based therapeutic approaches in the context of OVX mouse models for osteoporosis.

## Figures and Tables

**Figure 1 ijms-22-09069-f001:**
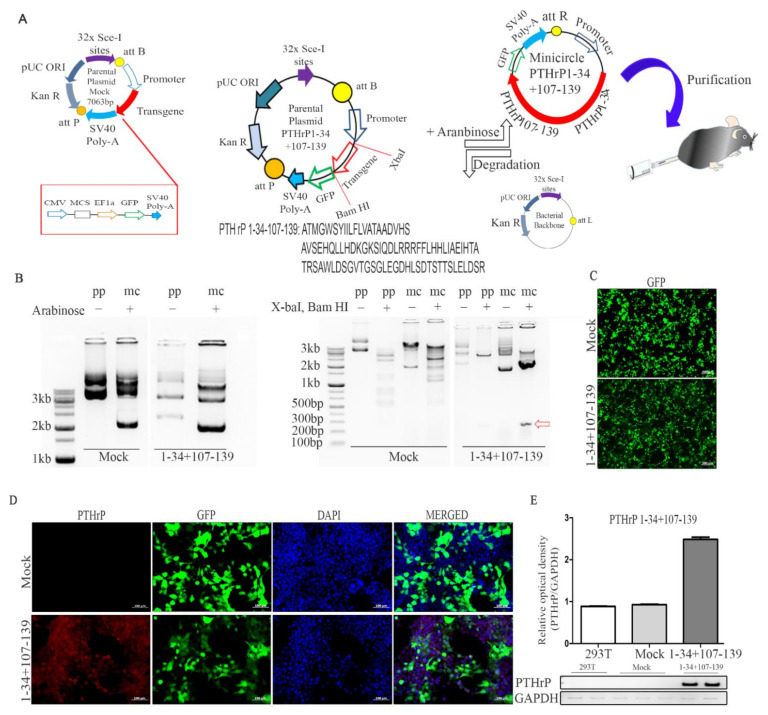
Production of the mc PTHrP 1–34+107–139 vector. (**A**) Schematics of the approach used for the generation of the mc PTHrP 1–34+107–139. (**B**) Representative agarose gel images of the parental (pp) and minicircle (mc) vectors and of the insertion confirmation after double digestion with restriction enzyme, XbaI, and Bam HI. (**C**) Fluorescence image of green fluorescence protein (GFP) expression in HEK 293T cells 48 h post-transfection with the mc PTHrP 1–34+107–139 vector; scale bar = 200 µm. (**D**) Immunofluorescence images showing the expression of PTHrP (red) and GFP in HEK 293T cells; scale bar = 100 µm. (**E**) Detection of the expression of PTHrP 1–34+107–139 (normalized to that of *GAPDH*) in a HEK 293T cell lysate, 48 h post-transfection.

**Figure 2 ijms-22-09069-f002:**
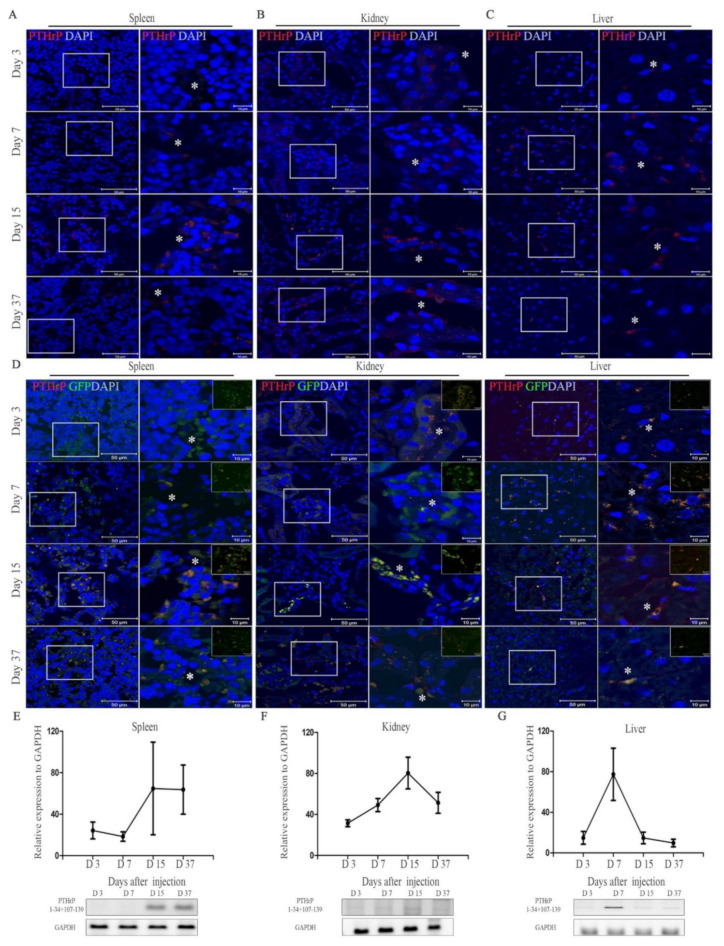
Expression of mcPTHrP 1–34+107–139 after in vivo injection. Immunofluorescence images of the detection of the PTHrP protein in the (**A**) spleen, (**B**) kidney, and (**C**) liver, 3, 7, 15, and 37 days after the intravenous injection of mc PTHrP 1–34+107–139 into the tail vein of mice; scale bar = 50 and 10 µm. (**D**) Immunofluorescence images of the co-localization of PTHrP and GFP in different tissues; scale bar = 10 µm. Relative expression of the PTHrP 1–34+107–139 gene in (**E**) spleen, (**F**) kidney, and (**G**) liver tissues collected 3, 7, 15 and 37 days post-injection. Results are given as the mean band intensity normalized to that of *GAPDH* (Image J software). White box indicates the magnified region of interest. A white asterisk indicates the merged region of interest.

**Figure 3 ijms-22-09069-f003:**
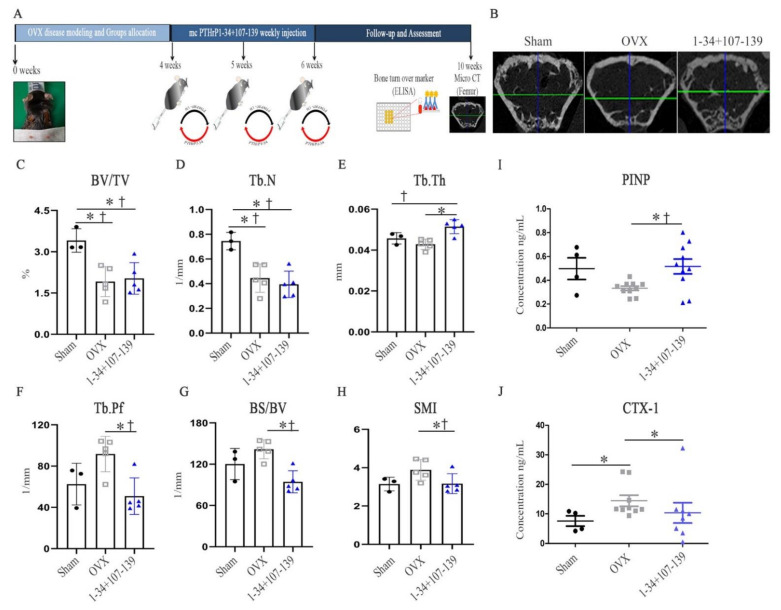
Bone-related parameters in mcPTHrP 1–34+107–139-administered versus control OVX mice. (**A**) Schematic diagram of the administration of mcPTHrP 1–34+107–139 to OVX mice; mcPTHrP 1–34+107–139 was administered intravenously three times (once weekly) from weeks 4–6 post-ovariectomy. (**B**) Representative micro-CT images of the femurs of the different groups of mice. The (**C**) regenerated trabecular bone volume fraction (BV/TV), (**D**) trabecular number (Tb.N), (**E**) trabecular bone thickness (Tb.Th), (**F**) trabecular bone pattern factor (Tb.Pf), (**G**) trabecular bone surface (BS/BV), and (**H**) structural model index (SMI) quality were determined in the femurs of each group. The levels of (**I**) PINP and (**J**) CTX-1 were also measured in serum samples collected from the different groups of mice, 10 weeks after surgery. The data are represented as the mean ± SEM. Statistical significance was assessed using the Kruskal–Wallis test with LSD (†) post hoc analysis and the Mann–Whitney (*) test: *† *p* < 0.05; n.s. = not significant.

**Figure 4 ijms-22-09069-f004:**
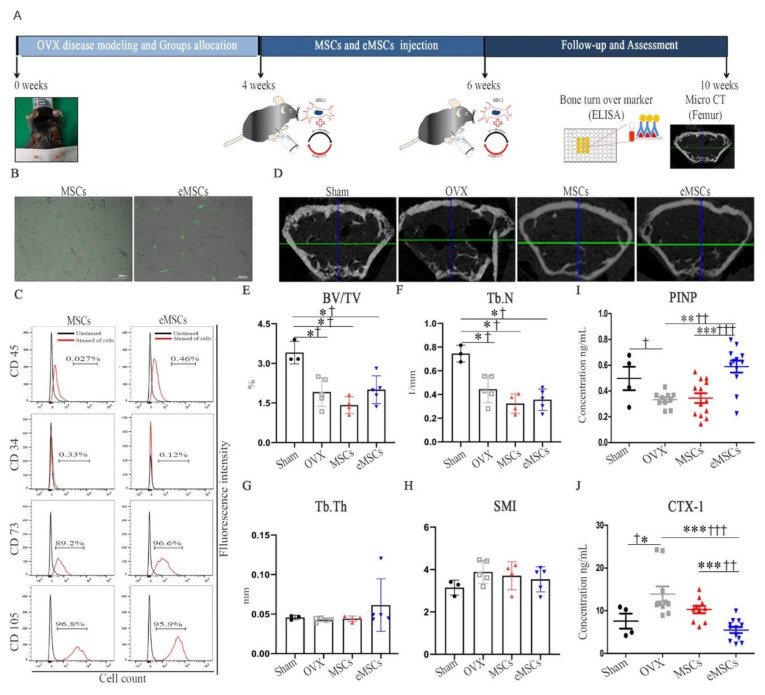
Therapeutic effect of eMSCs in OVX mice. (**A**) Experimental timeline. MSCs and eMSCs were injected twice at weeks 4 and 6 after ovariectomy, intraperitoneally. (**B**) Fluorescence images showing the expression of GFP 48 h post-transfection with mcPTHrP 1–34+107–139; scale bar = 200 µm. (**C**) MSC-specific marker expression in MSCs and eMSCs; black and red histograms represent unstained and stained cells, respectively. (**D**) Representative 2D micro-CT images of the femurs of the different groups of mice. The (**E**) regenerated trabecular bone volume fraction (BV/TV), (**F**) trabecular number (Tb.N), (**G**) trabecular bone thickness (Tb.Th) quality, and (**H**) trabecular bone structural model index (SMI) quality were determined in the femurs of each group. The levels of (**I**) PINP and (**J**) CTX-1 were also measured in serum samples collected from the different groups of mice, 10 weeks after surgery. The data are represented as the mean ± SEM. Statistical significance was assessed using the Kruskal–Wallis test with LSD (†) post hoc analysis and the Mann–Whitney (*) test: *† *p* < 0.05, **†† *p* < 0.01, ***††† *p* < 0.001, n.s. = not significant.

**Figure 5 ijms-22-09069-f005:**
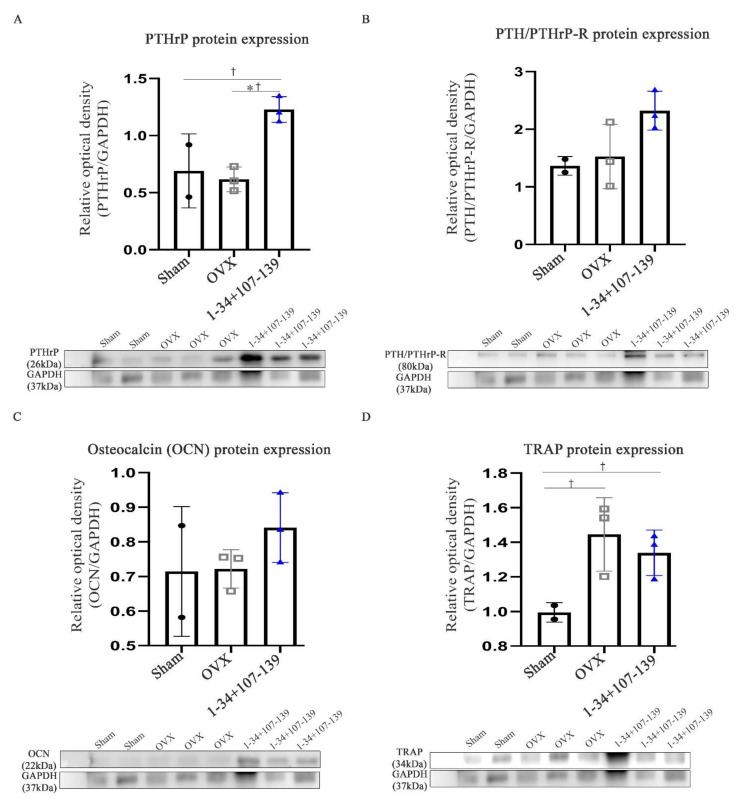
Protein expression of PTHrP, PTH/PTHrP-R, and bone formation and resorption markers in the femur after the injection of mcPTHrP 1–34+107–139 into OVX mice (10 weeks post-OVX). (**A**) Protein expression of PTHrP. (**B**) Protein expression of PTH/PTHrP-R. (**C**) Protein expression of osteocalcin (OCN). (**D**) Protein expression of TRAP. The data are represented as the mean ± SEM. Statistical significance was assessed using the Kruskal–Wallis test with LSD (†) post hoc analysis and the Mann–Whitney (*) test: *† *p* < 0.05, n.s. = not significant.

**Figure 6 ijms-22-09069-f006:**
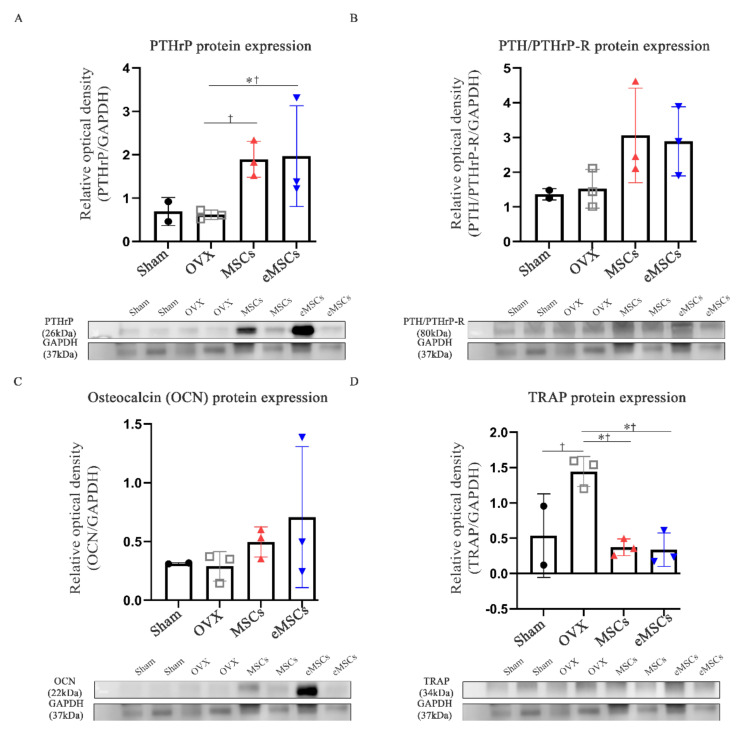
Protein expression of PTHrP, PTH/PTHrP-R, and bone formation and resorption markers in the femur after the injection of eMSCs into OVX mice (10 weeks post-OVX). (**A**) Protein expression of PTHrP. (**B**) Protein expression of PTH/PTHrP-R. (**C**) Protein expression of osteocalcin (OCN). (**D**) Protein expression of TRAP. The data are represented as the mean ± SEM. Statistical significance was assessed using the Kruskal–Wallis test with LSD (†) post hoc analysis and the Mann–Whitney (*) test: *† *p* < 0.05, n.s. = not significant.

## Data Availability

Data is contained within the article or Appendix A.

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
