# Peer review of "Increased Potential of Bone Formation with the Intravenous Injection of a Parathyroid Hormone-Related Protein Minicircle DNA Vector"

_ijms, 2021, doi:10.3390/ijms22169069_

Round 1

Reviewer 1 Report

All major deficiencies of the manuscript still remain and the manuscript can only be accepted if the authors show at least the following:

  1. Is there any effect of using minicircle DNA over plasmid DNA on bone formation? Using the ovariectomized mice alone can not serve as the appropriate control. The correct control will be the Parathyroid Hormone-related Protein plasmid DNA control. One needs to compare the effects of Parathyroid Hormone-related Protein when delivered using the minicricle DNA vector versus the plasmid DNA vector in order to understand if this technology has any positive influence compared to the other.

Without this one single control, the authors claim remain unanswered and can not be suitable to consider for publication.

Author Response

" Please see the attachment"

Reviewer 2 Report

The author tried to treat osteoporosis patients with mini vectors. This topic is interesting and it has visibility in gene therapy. However, there are still some issues that need to be resolved before the paper is accepted

Major question

  1. In Figure 2, the delivered mini vectors showed their strongest performance at Day 15 (Figure 2, A-D), and then its performance gradually weakened. it seemed that mini vectors could not last for a month?
  2. In Figure 2, Mini vector has different expression levels and time points in different organs (spleen, kidney, and liver)? How do explain these phenomena, Are there any relevant references?
  3. Although Mc PTHrP can enhance bone density, does Mc PTHrP affect other organs and cause side effects?
  4. Figure 5B, what is the difference between PTH/PTHrP-R and PTH/PTHrP-rR in the Y-axis
  5. Compared with abaloparatide and Mc PTHrp1-34+107-139, which one has the better benefit in osteoporosis treatment?

Minor question

  1. In Fig 1B, it is recommended to add an indicator for 1-34+107-139 bands between Xbal & Bam HI sites.
  2. Fig 2 figure legend (A) the space in the spleen should be cancel
  3. In Figure 6, the internal control of GAPDH should be used at the same concentration between samples to evaluate the expression of target proteins.

Author Response

" Please see the attachment"

Round 2

Reviewer 1 Report

I consider that the author's revision though not fully satisfied, the modifications made to understand the limitations and references provided for the acceptance of mc to deliver target genes in other tissues are acceptable. Therefore, I recommend the manuscript for publication.

Reviewer 2 Report

The authors had responded to previous recommendations and revised the manuscript properly.

This manuscript is a resubmission of an earlier submission. The following is a list of the peer review reports and author responses from that submission.

Round 1

Reviewer 1 Report

This is an interesting study investigating the potential of systemic engineered PTHrP delivery to increase bone formation in ovariectomised models of osteoporosis. While the study employs an innovative concept (including minicircle DNA) to deliver an anabolic factor via gene therapy, it has several shortcomings that need to be addressed.

Please see below for a detailed list of minor-major revisions that will need to be provided. Mostly these centre around lack of clarity, the vagueness of experimental detail, some issues with interpreting the results and minor language improvements.

I. Title:

The title states "...intravenous injection of parathyroid hormone-minicircle DNA". This is incorrect or in disagreement with the described methods and terminology throughout the manuscript. Which factor was delivered PTH or PTHrP? These are not the same.

II. Abstract:

"encoding the infusion of PTHrP" does not seem to make sense in this context. What is "infusion" referring to? The minicircle encodes the PTHrP factor.

IV. Introduction:

"The mechanism of anabolic agents increase bone formation" should read "increases bone formation".

The authors mention DKK1 and Sost as markers for osteoclasts, citing other work. This seems incorrect, as far as the reviewer is informed, these are markers for osteocytes.

The sentence "However, the anti-resorptive and bone anabolic agents showed side-effects and poor long-term efficacy for postmenopausal osteoporosis treatment." seems out of context. Which agents and how do they related to the previous statements?

V. Methods:

The description of the employed plasmid vector and contained elements is poor. It is not clear if the vector co-expressed GFP with the therapeutic PTHrP or if this is separate. There are rudimentary plasmid maps provided for the construct in Figure 1A, but these do not seem to make sense. Where is the PTHrP gene incorporated in the context of the map elements showing EF1a and GFP? Furthermore, Kozak is misspelt. Finally, the plasmid and construct maps' nomenclature has to correspond with the nomenclature used in the manuscript text body.

Gene expression studies: There is a lot of important information missing regarding the gene expression studies, particularly for the in vitro part. How was RNA transcribed to cDNA or was RT-PCR directly performed from RNA? Were transgene persistence studies performed by detecting pDNA itself or the specific transcripts expressed from the minicircle? It is essential to provide full details for these approaches! It is not possible to interpret the results presented due to a lack of methodological information.

Why were there no internal housekeeping genes measured to normalise the data in Figure 2 G?

Were OVX mice injected with naked minicircle pDNA or were any condensation/complexation or transfection reagents used to protect the circulating mc DNA from degradation/enhance cellular uptake? Please provide the total dose of DNA used per animal as mg/kg body weight.

The main rationale for the current work seems to be treating bone mineral density (BMD) and bone mass reduction in osteoporosis using an anabolic agent. Why was there no assessment of BMD differences then?

Histological procedures and stainings are not described in sufficient detail. How was GFP fluorescence preserved during fixation and histological processing?

VI. Results:

"To investigate the effects of long minicircle affected cells..." - what is a long minicircle?

The authors state on page 8 that "Interestingly, the expression of the PTHrP antibody...". This seems incorrect as you do express PTHrP and not the antibody used for detection.

V. Discussion:

What is the definition of "quality of trabecular bone structure"? The study would have benefited from correlating to a clinically used outcome parameter for bone quality or BMD used for osteoporotic patients.

Reviewer 2 Report

Increased Potential of Bone Formation with Intravenous Injection of Parathyroid Hormone-Minicircle DNA Vector.

This manuscript uses the novel gene delivery system minicircle DNA vector to deliver the PTHrP gene a previously well-known protein secreted by a subset of chondrogenic/skeletal stem/progenitor cells that is essential to maintain skeletal homeostasis. The novelty of this manuscript entirely lies in the demonstration of the efficiency of the delivery system, if possible target specificity should also be a goal. However, the authors aimed to deliver the gene by injecting intravenously into the experimental mice, and therefore it is understandable that the focus on achieving bone-specific expression of PTHRP selectively in bone alone is not possible. Given that the anabolic/bone sparing effects in ovariectomized mice is clearly evident, it is easy to speculate that the expression or availability of PTHRP at the bone site could be sufficiently high enough to induce the observed PTHRP-mediated effects in bone. 

However, the authors went further to inject the PTHRP gene-modified MSCs intraperitoneally and in a similar fashion show effects on bone formation/resorption without demonstrating the presence of transgene or the incorporation of newly injected cells into the target bone tissues. It is essential that the authors demonstrate that the newly introduced PTHRP transgene is available at the bone site. The authors are required to show either the bone tissue or marrow or harvested stromal cells from injected mice also express the transgene as is demonstrated for spleen or kidney or liver in the current manuscript. Alternatively, they may otherwise show that the sections of the bone tissue harvested from mice after injection exhibit the transplanted MSCs (if they have injected any form of labeled MSCs).

This is essential to delineate whether the PTHRP_MSCs harboring the bone are directly involved in the bone remodeling effects or if the effects are derived from the high levels of PTHRP protein (not the MSCs) alone at the bone site. This could further explain whether the effects are from the exogenous PTHRP-modified MSC-derived or is derived from increased production of PTHRP from the PTHRP_MSCs that harbor the neighboring tissues/sites (NOT BONE) or from the PTHRP expressing tissues such as the spleen, liver, and kidney that secrete large amounts of PTHRP protein (shown in figure 2) that may enter bone through circulation and not from the bone itself if the authors do not see that kind of expression of the transgene in the bone.

Other than this one concern, the manuscript does not have any deficiency and can be considered for publication after establishing whether there is increased expression of PTHRP (or) NOT in the bone compartment.

Round 2

Reviewer 1 Report

The authors have addressed most of the comments satisfactorily; however, there are still several major concerns re. study details that need to be answered/addressed. Furthermore, the revised version contains several typos. Therefore, I would suggest to proofread it again meticulously.

Major concerns re. responses:

  1. From the more detailed plasmid map layouts provided in the revised version, it seems as if the therapeutic mc also contains the GFP gene downstream of the PTHrP gene. If so, has this been cloned to be inframe with GFP to obtain the expression of a PTHrP fusion gene? The reviewer assumes that this is the case as GFP detection is used in the therapeutic group to prove mc expression and co-localisation with an immunofluorescent PTHrP signal. Therefore, the reviewer questions why the authors have deliberately (in the initial submission and the revision) omitted to describe this important aspect in detail. If the expressed protein from the therapeutic mc is a PHTrP-GFP fusion protein, then this needs to be stated throughout the manuscript as well as in the title, and it is questionable if PTHrP bioactivity of such a fusion gene is maintained.
  2. RT-PCR experiments: The authors provide essential information on the RT-PCR experiments performed to determine PTHrP transgene expression in vitro and in vivo. How was the measurement of transcript-derived signals ensured as low molecular weight mc DNA would co-purify with RNA during extraction and would give false-positive signals? Were RT-negative controls performed (no reverse transcriptase = no cDNA transcription) to determine if there is a residual signal from the plasmid (which is not associated with successful gene expression)? The authors furthermore provide more details regarding the in vivo gene expression experiments: Is it correct to assume that there was no intermediary cDNA synthesis step (as in the in vitro experiments) and this RT-PCR was performed as a one-step RT-PCR experiment? If so, which kits and enzymes were used?
  3. Histology: Was GFP histology performed using cryo-sectioning and was native GFP fluorescence detected in samples rather than using an antibody? If so, please comment on the necessity of using specific methods to preserve GFP fluorescent signals. In the reviewer's experience, GFP signals tend to be lost due to fixation on PFA, which was used in the current study. 
  4. Please confirm that all PCR data presented refers to RT-PCR and was intended to measure the transgene transcript and not the plasmid DNA persistence.
  5. The abbreviation "mc" in the abstract needs to be defined.

Reviewer 2 Report

I have no more concerns.

ACCEPT.

Round 3

Reviewer 1 Report

Important study design details have been omitted from the initial submission and now been provided after significant probing by the reviewer, indicating that what was reported initially was not an accurate representation of the materials used and experiments performed. Even if this major flaw was ignored it is questionable how robust the current study design is (expression of a PTHrP-GFP fusion protein, no data on bioactivity; potential false-positive signals from qPCR as no controls performed for vector contamination). Unfortunately, this is not sufficient quality for publication.